# A Strategy for Enhancing English Learning Achievement, Based on the Eye-Tracking Technology with Self-Regulated Learning

**Yu-Chen Kuo [1], Ching-Bang Yao [2],\* and Chen-Yu Wu [1]**

1    Department of Computer Science and Information Management, Soochow University, Taipei 100006, Taiwan
2    Department of Information Management, Chinese Culture University, Taipei 11114, Taiwan
\*    Correspondence: yqb@faculty.pccu.edu.tw or chinbang7@yahoo.com.tw

**Abstract:** Owing to the global promotion of e-learning, combining recognition technology to facilitate learning has become a popular research topic. This study uses eye-tracking to analyze students' actual learning situations by examining their attention during the learning process and to provide timely support to enhance their learning performance. Using cognitive technology, this study can analyze students' real-time learning status, which can be utilized to provide timely learning reminders that help them achieve their self-defined learning goals and to effectively enhance their interest and performance. Accordingly, we designed a self-regulated learning (SRL) mechanism, based on eye-tracking technology, combined with online marking and note-taking functions. The mechanism can aid students in maintaining a better reading state, thereby enhancing their learning performance. This study explores students' learning outcomes, motivation, self-efficacy, learning anxiety, and performance. The experimental results show that students who used the SRL mechanism exhibited a greater learning performance than those who did not use it. Similarly, SRL mechanisms could potentially improve students' learning motivation and self-efficacy, as well as increase their learning attention. Moreover, SRL mechanisms reduce students' perplexities and learning anxieties, thereby enhancing their reading-learning performance to achieve an educational sustainability by providing a better e-learning environment.

**Keywords:** eye-tracking; learning motivation; emotion recognition technology; learning attention; learning anxiety; educational sustainability

## 1. Introduction

Reading e-books and using other digital reading tools have become common practice today. Such activities enable students to acquire a second foreign language via the Internet, helping them to understand the content quickly and effectively. Online learning requires students to be able to acquire knowledge on their own; thus, their self-regulated learning (SRL) ability affects their reading and learning performance [1]. Several studies have developed reading systems or devices that aid students in reading, which indicates the tremendous extent to which online reading has become popular in the field of e-learning [2, 3]. Attention is one of the factors that affects the learning performance during the reading process. Studies have revealed a significant correlation between learning attention and learning outcomes [4,5]. Therefore, encouraging students to focus on the material is a challenge confronting digital learning. Even when students encounter simple or difficult learning content, how to properly help students not to be distracted and to continue to focus on the content they are learning is one of the main motivations of this study.

Many scholars have actively discussed the combination of cognitive technologies to aid e-learning. Cognitive technology uses sensors to detect learning situations, such that timely help can be provided to enhance the learning performance of students. Analyzing their attention during the learning process is an important aspect in the application of cognitive technology. Research has indicated that using a cognitive technology system is helpful

in detecting the students' learning status and helping them learn better [6]. Cognitive technology can, therefore, be used to examine the students' learning status, assist them in completing their learning goals, and help them build their own knowledge structure [7,8].

Cognitive technology has primarily been used to discuss the interaction between students and their learning environment. It can also be applied to identify learning emotions, according to which learning feedback models can be designed to adjust the teaching process. If students are uninvolved in learning activities, their learning outcomes will be unsatisfactory, even if a high-quality learning environment and content are provided to them. Therefore, in recent times, emotional cognitive technology has begun to receive scholarly attention. For example, a study [9] detected students' emotions and established an interactive learning environment, based on emotional cognition. Emotions can directly reflect a student's current learning situation. Therefore, their learning emotions and attention should be detected using a combination of physiological sensing devices. An assistive learning mechanism, designed to provide real-time feedback during the learning process, would be significantly effective for this purpose.

Additionally, studies have indicated that students who do not have the ability to adjust their self-regulated strategies in the course of the e-learning process will be unable to attain an in-depth understanding of complex topics [10,11]. Some have also suggested that if students do not use effective metacognitive strategies for self-adjustment during hypermedia-based learning processes, they will fail to obtain knowledge effectively [12]. In other words, students' SRL abilities are crucial to achieving a good learning performance in the e-learning process. This is the second main research motivation of this study.

Moreover, with regard to learning foreign languages or a second language (L2), Cohen [13] mentioned that, in the L2 learning process, although the individual characteristics of the students greatly affect the speed of learning a second or a foreign language, learning methods, strategies, and motivation to learn, are key factors that teachers can actively address to improve their teaching efficiency [13]. These important principles of foreign language learning also apply to e-learning.

This study primarily aims to develop an English SRL mechanism, integrated with eye-tracking technology for English reading. This mechanism would assist students' SRL and provide them with tools to enhance their learning performance. In this system, students set their own learning goals before they begin. During the learning process, students are provided with a marking and note-taking function that is different from that provided by a conventional learning system. The SRL learning system developed in this study is designed with a marking and note-taking learning module, which is aimed at targeting the learning function of the 5W1H (who, when, where, what, why, and how) reading method commonly used in English teaching. This, in turn, guides them to pay attention to the structure of the article, assisting them in SRL. With the above research motivations, the main purposes of this study are to provide a better e-learning platform with the SRL mechanism to enhance the effectiveness and efficiency of e-learning. English learning is used as the experimental field.

The eye tracker enables the system to record the eye movements of students, effectively, providing learning assistance and feedback to allow the students to return to a better learning state, thereby improving their SRL performance. Once they have learned, the system presents various SRL statuses, including learning achievement points, reading attention points, and gaze points, based on the thermal images of eye movements, which can enable the students to rethink their SRL status and enhance their reading ability. This study explored students' learning performance, motivation, self-efficacy, and learning anxiety, to examine the effectiveness of the SRL mechanisms. Additionally, we further explored how students change their status during the learning process.

Accordingly, an experiment was conducted in a university to explore the effectiveness of the proposed approach. The following research questions were examined:

1. Can the SRL mechanism with eye-tracking technology improve the students' learning performance in English reading courses?

2. Does it improve the learning performance in English reading courses for students of different genders?
3. Does it improve students' attention (or reduce their confusion) while they are learning from English reading courses?
4. Does their learning attention (or confusion) and learning performance exhibit significant positive correlations in the case of English reading courses?
5. Can the SRL mechanism with eye-tracking technology improve students' learning motivation and self-efficacy, and reduce learning anxiety while learning from English reading courses?

## 2. Related Research and Applications

### 2.1. English Reading

English is the most widely spoken language in the world, and in most countries, is considered the most important foreign language to acquire. In non-English-speaking countries, many people are striving to find efficient learning models or tools to enhance their English proficiency. In traditional English language courses, reading is considered the most important skill [13].

Reading is a complex psychological process. Prior to the readers understanding of the meaning of an article's content, they use their prior knowledge to guess the meaning of the content; then, through continuous circulation, they improve their understanding of the meaning [14]. Therefore, reading necessitates not only the students' reading skills, but also their understanding of the implied themes between the sentences and paragraphs, as well as their ability to evaluate various viewpoints. Hence, the reading process involves background knowledge and reading monitoring [15].

Recent studies on English reading have focused on the importance of reading strategies. Students should make good use of reading strategies to plan how to read and enhance their reading performance [16]. This indicates that designing a fluent reading mechanism can effectively improve the learning outcomes in English reading. Therefore, to enhance the students' English reading performance and motivation, it is important to combine the reading strategies and activities, develop aids to support the students' English reading, and reduce students' anxiety.

### 2.2. Reading Strategy

A study used a self-questioning strategy to allow students to review their learning [17]. It found that students' self-questioning after learning can effectively improve not only their English reading ability, but also their capacity to think.

Studies have confirmed that reading strategies can help students read effectively, improve their learning performance and reduce their learning anxiety. These methods include marking and answering questions. The marking strategy has been widely applied in reading activities during teaching. Studies have found that using markings during foreign language reading can effectively help in learning [18]. Scholars have adopted different marking strategies. For example, a study [19] proposed four notes, including summary, paraphrase, verbatim, and words beginning with capital letters; the results showed that the marking content and the depth of the notes significantly affect reading.

Therefore, in this study, we considered these reading strategies, to design a self-regulated English learning system that can help students use these strategies for learning. Students could set their preferred learning strategies, based on the results of previous learning, such as learning time and chapters. They followed our adopted reading strategies through the marking and note-taking learning module of our SRL system, based on the famous 5W1H strategy: character (who), event time (when), location (where), article event (what), purpose of the article (why), and article's conclusion (how). This strategy helped students learn the article structure, which further helped them understand the text. Students reviewed their learning status and the system further analyzed their attention and learning time, and provided a thermal image of their gaze on the material to achieve

a monitoring strategy. In the case of students who were confused, following the test, we also highlighted the key sentence, guided them to re-read the key passages of the article, and helped them undertake correction strategies. The adoption of these strategies can help reduce the learning anxiety of students and help them achieve a better learning effect, thereby promoting the sustainable education goal of digital learning [20].

*2.3. Self-Regulated Learning*

Bandura proposed the concept of SRL in 1977, revealing that students can establish and maintain their learning motivation through goal setting, self-evaluation, and self-enhancement. Helping students adjust their own learning status in SRL, based on previous learning experience, reinforces the lack of the next study, which is the goal of teaching.

To encourage students to proactively plan their own learning, scholars have defined SRL as a learning method that includes goal setting, strategy use, self-monitoring, and self-adjustment. Accordingly, they proposed a SRL framework [21,22]. Students with high learning achievements set clear goals for their own learning [23–25]. Moreover, such students use more strategies to help themselves in the learning process. They also monitor their learning processes more frequently and follow up with the learning results. Previous studies proposed a SRL cycle model that defines SRL as a learning method [26]. It includes four steps: self-evaluation and monitoring, goal setting and strategy planning, strategy implementation and monitoring, and strategy outcome monitoring.

The cycle model of SRL helps students self-observe and self-evaluate, set goals, use learning strategies, and monitor their learning process. It also allows them to self-reflect and adjust their own learning methods, to enhance their performance. Furthermore, SRL strategies can enhance students' willingness to learn [27].

Some scholars in the past proposed a conceptual framework for SRL (Table 1). In this study [28], we designed a SRL system, based on this learning concept. This system allows students to customize their learning goals to correspond to the "Why" and provides them with the marking system to be used while reading, to correspond to the "How". Furthermore, students can identify their learning time through the system, to examine their learning performance, which corresponds to the "When", in the learning framework. Additionally, it provides them with a learning status and history to review their learning performance and make necessary corrections that correspond to the "What" in the framework.

**Table 1.** SRL conceptual framework.

| Topic | Level | Student Condition | Self-Regulated Attribute | Self-Regulated Process |
|---|---|---|---|---|
| Why | Motivation | Choose to participate | Self-motivated | Self-efficacy and self-goal |
| How | Method | Choose method | Planned or automated | Strategy use |
| When | Time | Choose time limit | Timeliness and efficiency | Time management |
| What | Behavior | Choose learning result | Self-awareness of performance | Self-observation, self-judgment, self-reaction |
| Where | Environment | Choose environment | Environmental sensitivity and strategy | The construction of the environment |
| With whom | Social | Choose peer, model, or teacher | Socially sensitivity and strategy | Selective seeking of help |

However, based on the related work discussed above, most studies related to SRL are primarily focused on the role of non-synchronized online learning environments, and they assist traditional teaching, study of the SRL ability, according to the learning process, or use of the Likert or the MSLQ scales. Open questionnaires have been used to conduct research on SRL. Currently, few studies have developed online learning systems that can monitor and detect students' SRL performance. Therefore, this study intends to use eye-tracking technology to help readers pay attention to learning materials. Using this technology, students can monitor and detect their SRL performance in real time. If the students do not pay attention or are confused, the system can immediately provide assistance, helping them to return to the learning state, enhance their attention, reestablish a better learning mood, and continue with SRL.

### 2.4. Learning Attention

There are various definitions of attention. This study measured attention using eye-tracking technology [23]. Attention refers to the level of concentration and focuses on specific affairs while performing a task or thinking about effectively dealing with a specific issue [29,30].

Studies suggest that students' behaviors affect their learning and academic performance, and attention is the key indicator of these factors [31–34]. Throughout the learning process, in both traditional and digital approaches, students' attention is an important factor [35–37]. During online learning activities, greater attention has led to a higher learning performance [38]. Therefore, sustaining students' attention during the learning process is essential.

Most previous studies have used brainwaves to gauge the readers' attention. A set of students' brainwave attention monitoring reminder systems was established to enhance the learning performance [39]. However, brainwave recognition technology can only sense attention; it cannot determine whether students are concentrating on the learning materials or on something else. In other words, students' brainwave attention may remain high even when their sight is not engaged in reading. Studies have investigated learning in three types of materials and found that the group with the highest brainwave attention exhibited the worst learning outcomes [40]. In other words, brainwave recognition technology unilaterally considers the brainwave values while ignoring the limitations that students should consider in materials. To overcome this limitation, we used eye trackers to assist students in focusing on their learning materials. Using eye-tracking technology, students can monitor and detect their SRL performance in real time.

### 2.5. Emotion Recognition Technology and Eye-Tracking Technology

Emotions can affect students' memory and learning performance, and teachers can monitor their students' learning states with a computer and adjust their teaching strategies accordingly [41,42]. Since the maturation of emotion recognition technology, scholars have begun observing students' responses to emotions in the field of digital learning.

The E-learning uses different media materials and uses physiological sensors. Understanding students' emotional reactions can help ascertain the impact of various media on their learning performance. Additionally, a study explored the impact of anxiety on students' learning outcomes. Their experiment used personalized English teaching via distance education, as a goal, and they provided a sensor to detect the students' emotions while they were learning. When the student faced any difficulty and exhibited an anxious emotional state, the learning system informed the teacher of the need to use appropriate teaching strategies to address the student's emotions and to help them learn better. Attention to learning emotions has been regarded as a key factor in learning. However, it is limited by lack of research in the development of attention detection technology. Recently, owing to the development of physiological signal sensing technologies, existing companies have developed attention-sensing tools.

As early as 1879, the development of eye-movement technology was recorded in an observational manner, with patterns, such as fixation and saccade, being observed. An eye-tracking device was first developed in 1908. In the beginning, an invasive eye-moving device was used. It was a contact lens-like device, made of gypsum plaster, placed on the subject's cornea and attached to an aluminum bar. The device measured and recorded slight tremors of the subject's eyeball and the situation in which it beat slightly. Scholars indicate that the development of eye-movement technology has traversed three stages. The first stage occurred between 1879 and 1920. The initial research primarily focused on observations, it possessed a handful of instrumental records, and developed a less invasive observation device. In 1922, the related research focused on the eyeball and recorded the reflected light, collecting numerous values for eye-tracking data; it found that reading behaviors are more divergent and may differ, owing to differences in textbooks and purposes. At the beginning of the 20th century, since it was considered a stage of behaviorism, eye-movement research entered the second stage. This period advocated the exploration of psychology in a purely experimental mode and increased the significance of the combination of eye-movement research and psychology. In the mid-1970s, the decline in cognitive psychology led to the re-appreciation of the related research by scholars, and due to advances in science and technology, eye-tracking data became easier to obtain and analyze, hence, related research also appeared soon after.

Many studies have highlighted the relationship between eye movements, the learning performance, and attention. Research on gaze is mainly aimed at exploring the level of attention of the participants. Even if different subjects are studying the same topic, when the gaze time is longer in the right place, the attention paid by the student is considerably higher, as is the concept-building score associated with the topic. The results thus obtained indicate that the distribution of attention is related to the conceptual construction. Scholars, such as Sun et al., found that eye movement and physiological feedback show the users' relevance and difficulty in playing a game. The research focused on the saccade path primarily explored the puzzled mentality of the subjects or on their search for data. The study indicated that when readers were confused about a sentence, they read it again and searched for the required information by regression. However, few studies have combined real-time eye movement detection and self-disciplined learning, to determine and improve the concentration of students in real time, and to influence and improve the effectiveness of learning. Therefore, this study mainly focuses on two aspects. The first is to introduce the self-regulated learning mechanism, which is one of the most important points of e-learning. That is: how to arouse students' interest in learning through the learning functions provided by the e-learning platform, and then be able to actively learn and explore the learning contents. Another focus is on the development of the e-learning function using the eye-tracking detection technology, in order to help e-earning students to better solve the difficulties in the learning process, and then effectively improve their learning concentration.

## 3. Intelligent Personalized Context-Aware Recommendation M-Learning System Framework

This study primarily aims to provide an effective way of e-learning and use the online note-taking learning functions provided by the e-learning system. This system was developed to guide students to follow efficient reading learning strategies, thereby reducing the learning anxiety and load, and improving the learning motivation. This study uses the e-learning mechanism with SRL and eye tracking technology, to detect and understand the learning state of the students in real time (whether they are focused), and provide the digital learning function, such as the online marking and note-taking function, to assist students in following an effective 5W1H reading strategy, to conduct self-managed learning.

### 3.1. Conceptual Framework

To understand the feasibility of the SRL mechanism in English reading, this study adopts different learning mechanisms to explore its effects on English reading. These effects include the learning performance, learning motivation, self-efficacy, learning anxiety, and the degree of attention and confusion. The conceptual framework is illustrated in Figure 1. The independent variables are the different learning mechanisms and genders. The learning mechanisms include the use of SRL systems and general multimedia-learning systems; the gender variables are male and female. The covariate is the student's prior knowledge. The dependent variables include the learning activities and the effects of the learning performance, motivation, anxiety, self-efficacy, attention, and confusion during English reading. Regarding the gender factor, whether male or female, when using e-learning platforms for English learning, basically, the attentions or concentrations on the language learning is normally distributed, so there is no gender difference. That is to say, the learning concentration is related to the students' interest and learning performance. Especially in the subject of English learning, gender differences have no effect on students' differences in language learning. At the same time, our experiment also confirmed that there is no significant difference in the learning concentration for gender.

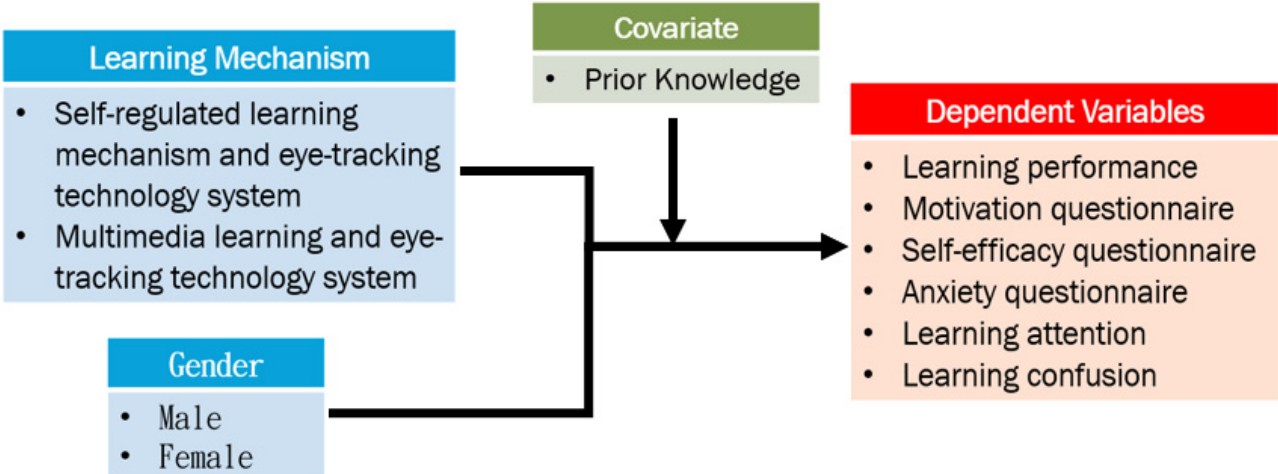

**Figure 1.** Conceptual framework and mechanism in the self-regulated learning platform.

### 3.2. Learning Environment

This study utilizes eye-tracking technology to develop an English reading SRL system, based on the importance of attention and the self-regulated ability in reading and learning. Using this technology, the students' fixation on the material while learning was identified, and their real-time learning mood was estimated using the sensed information, so timely help could be provided when they could not focus on the material. Emotions can directly reflect the students' current feelings. Therefore, if a good learning environment is ensured and effective learning strategies are implemented, it can stimulate the students' good learning emotions, and allow them to achieve SRL. Therefore, the SRL e-learning system with eye-tracking technology in this study can effectively improve the learning performance [43].

The SRL environment integrated with eye-tracking technology is shown in Figure 2. Following the calibration of the eye tracker in the computer classroom, the students could set their own learning goals and subsequently read the e-learning materials stored in the database. In "Set learning goals" area is the interface displayed in Chinese, which is mainly for students to fill in the estimated learning time, learning concentration, content perplexity, and test scores. The information in this interface shows that students can Check their own learning results after e-learning, which is also the main spirit of the SRL learning mechanism. In the learning process, the eye tracker senses and records the eye movement of the students and then transmits the eye movement learning record to a distance learning database for storage. Once the lesson is over, the system provides the indicators of the students' SRL; thus, students can self-examine their SRL status, to improve their performance.

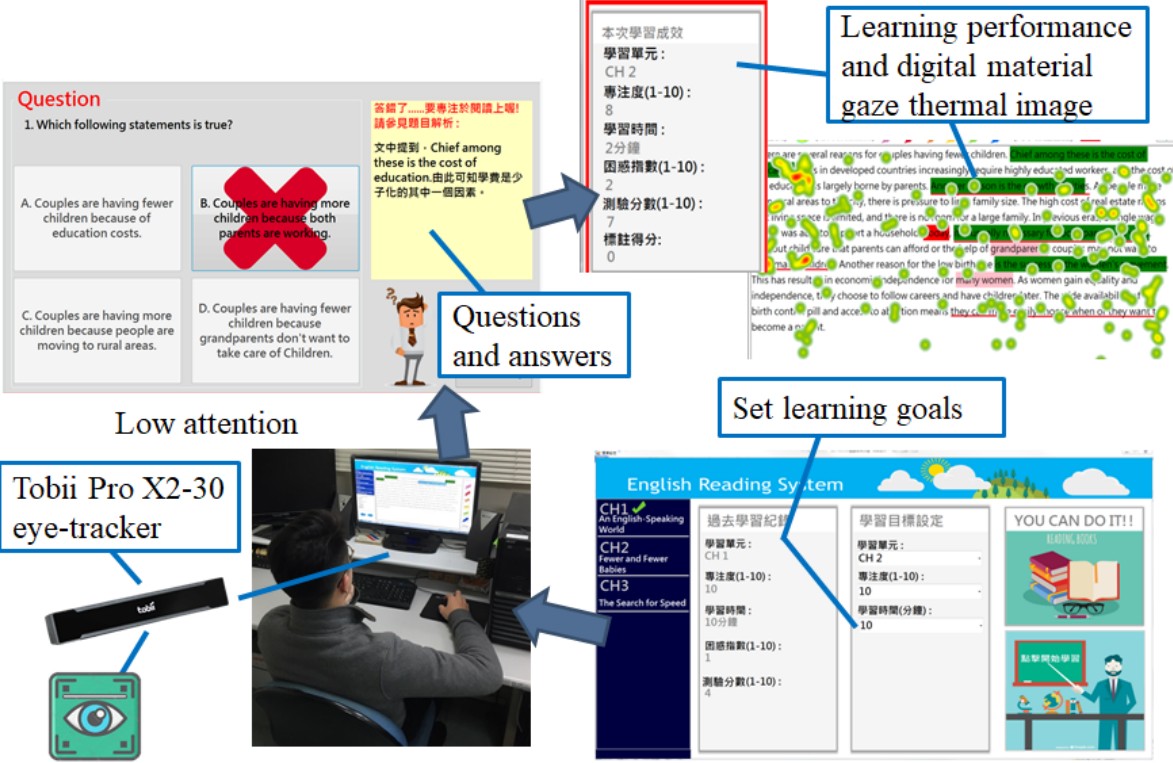

**Figure 2.** Eye-tracking English learning environment in the self-regulated learning platform.

### 3.3. System Interface

Integrating the English reading SRL system and eye-tracking technology involves the following functions: Tobii eye-tracking, the SRL goal-setting form, the learning emotion reminder, marking and note-taking, the quiz, the review, the SRL performance-checking, and one that determines whether the eye tracker performs the detection.

(i) Tobii eye-tracking function: It is one of the core functions. The system provides the corresponding auxiliary strategies, based on the detected eye-tracking information. Students can use the system to learn in English reading courses; when they gaze at the materials, their eye gaze data is detected and the learning strategies are provided to them, based on their status.

(ii) SRL goal-setting: Students can set learning goals, as shown in "student's target" in Figures 3 and 4, which is based on the best learning practices of past students. The set indicators include the learning time, learning units, and attention level. The system uses the learning goals set by the students as measurement indicators for the learning performance, and includes confusion and test scores (Figure 3) [43].

(iii) Marking function: The SRL e-learning system developed in this study has a marking and note-taking learning module; thus, e-students can make systematic key marks and notes immediately for the important or unclear content in the English reading process. When students are learning, they can use the article structure marking function, provided by the system, to determine the 5W1H reading strategy, marking them with different color markers and taking notes. This helps them focus on the article structure and understand the ideas in the article, which can further improve their reading performance and achieve SRL (Figure 4).

(iv) Quiz function: At the end of each paragraph, a quiz is provided for the students to examine their learning outcomes, to stimulate them to focus on the content of the material, to assist their refinement strategies, and to provide answers for the instant feedback (Figures 5 and 6).

(v) SRL performance-checking function: Once the reading is completed, the system displays the student's learning results on the system outcomes analysis page, including learning chapters, learning time, attention level, quiz scores, thermal images of the reading gaze, and marks between the teachers and students. This study uses the dwelling time of the student's gaze as the basis for measuring the attention level, as shown in Figure 7 below.

The intelligent e-learning system provided in this study uses eye-tracking technology to detect real-time images of digital students, to analyze the concentration of the students, in real time. Although it is impossible to instantly detect and understand the learning status of the students from a psychological viewpoint, it is possible as an outside observation. Nonetheless, the system can still obtain a certain degree of personal learning characteristics of e-learning students and provide personalized learning reminders to the students to facilitate more effective learning, which is based on the student characteristics emphasized by Cohen [13].

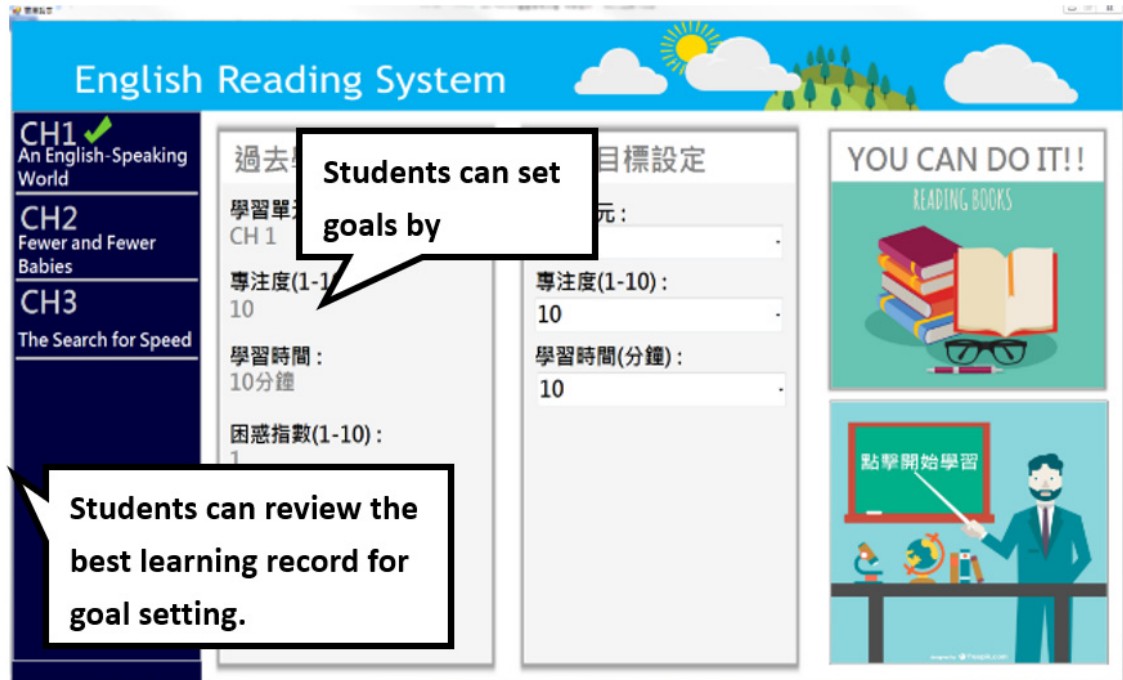

**Figure 3.** SRL goal-setting form in the self-regulated learning platform with eye-tracking technology.

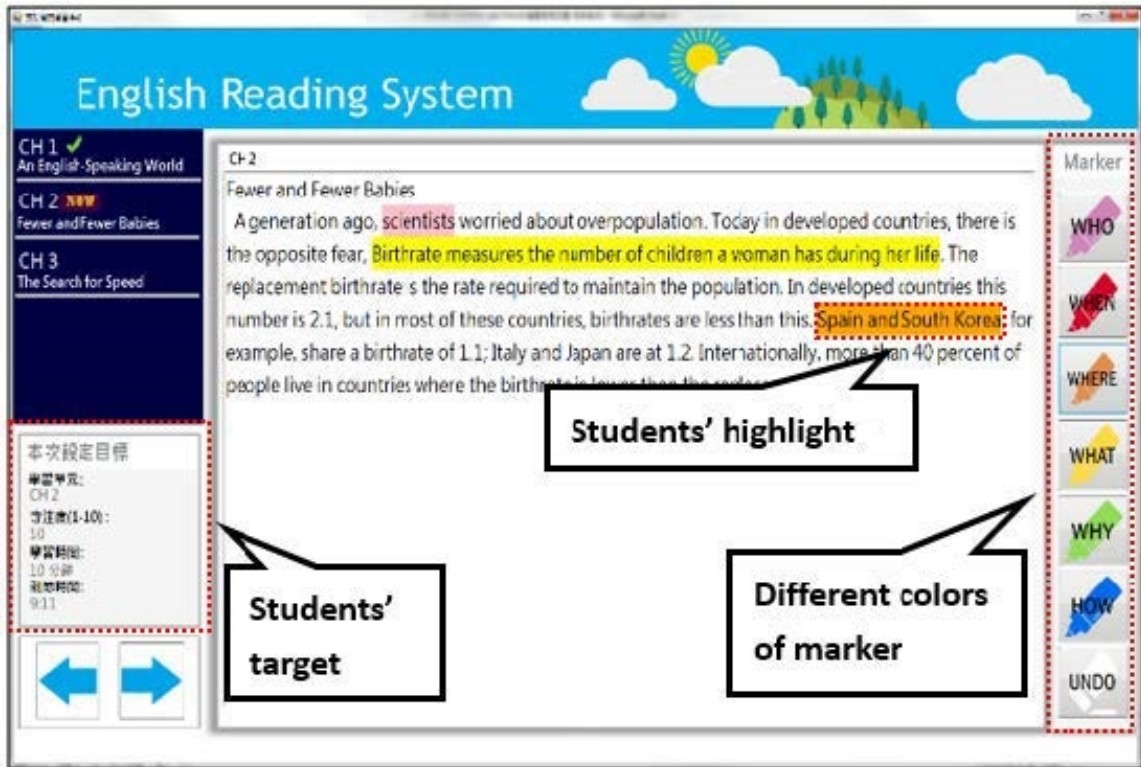

**Figure 4.** Marking function in the self-regulated learning platform with eye-tracking technology.

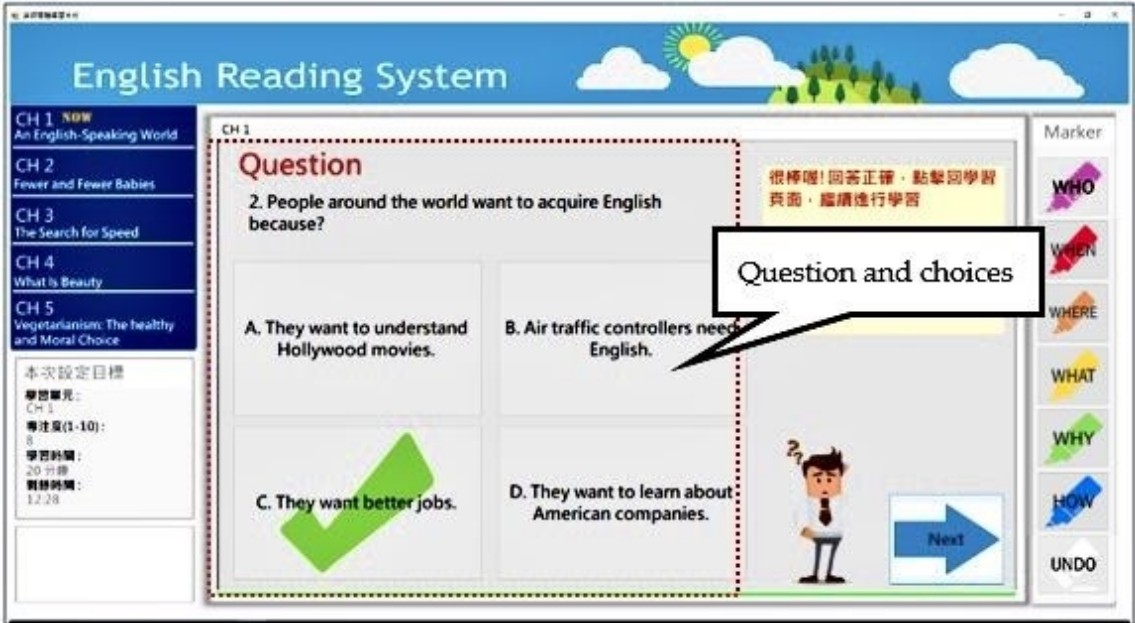

**Figure 5.** Quiz function and correct answers in the self-regulated learning platform with eye-tracking technology.

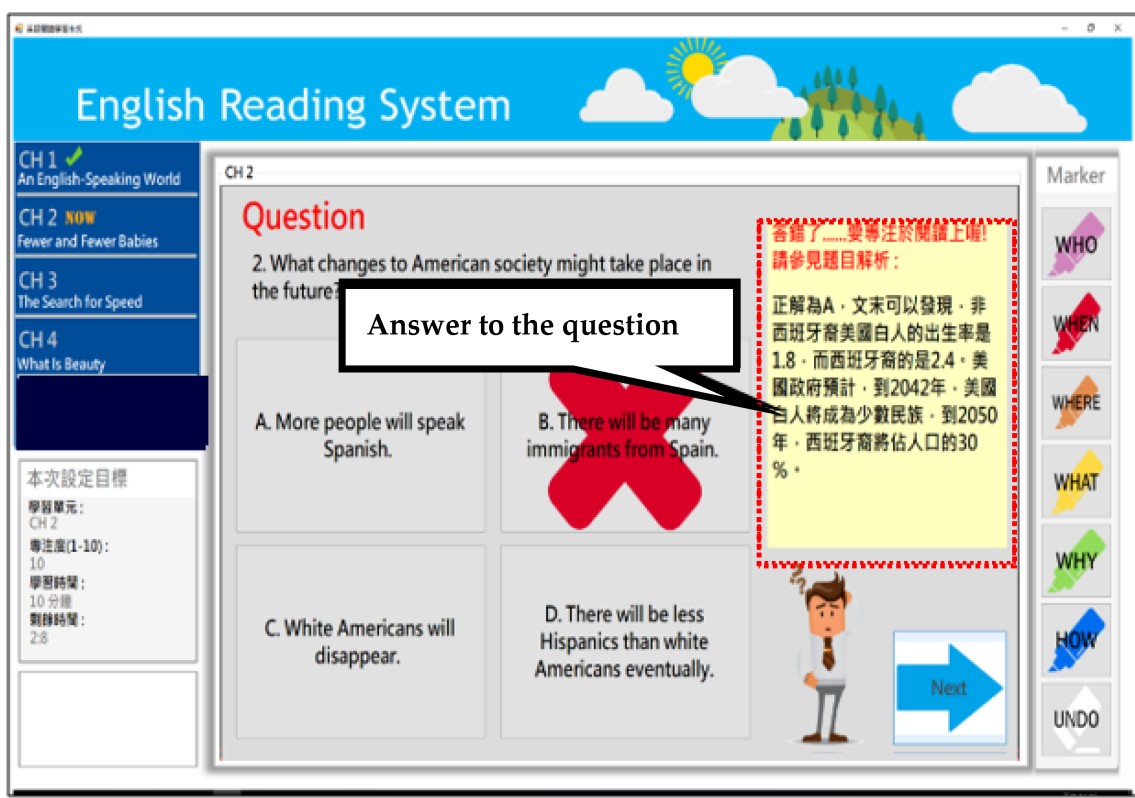

**Figure 6.** Quiz function and wrong answers in the self-regulated learning platform with eye-tracking technology.

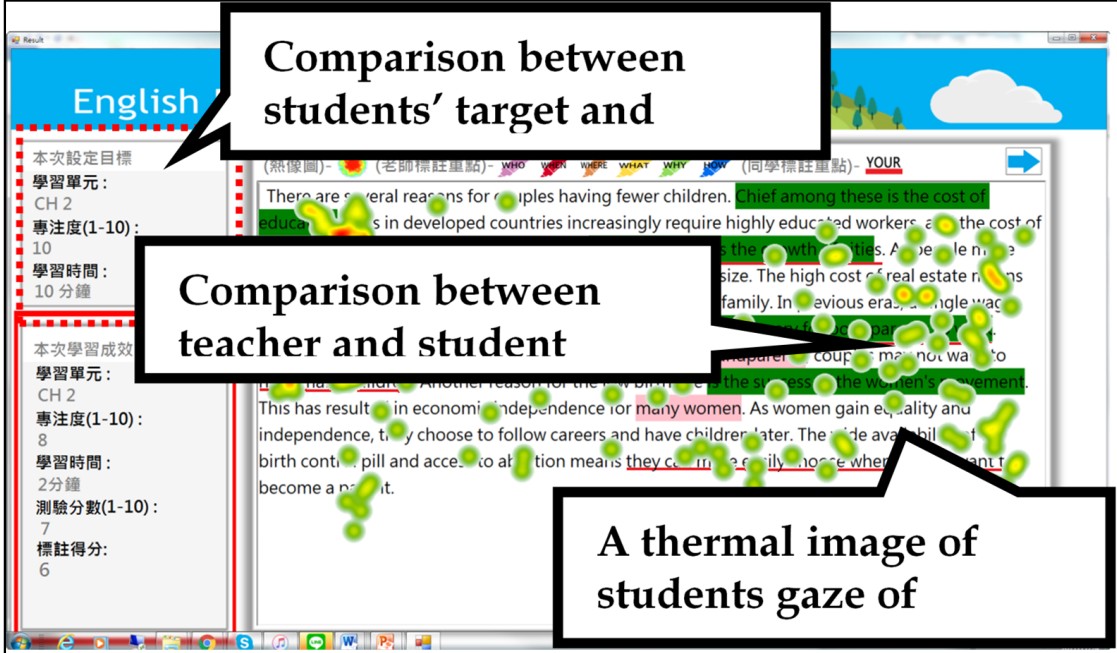

**Figure 7.** Self-regulated learning performance checking function in the self-regulated learning platform.

## 4. Experimental Design and Results Analysis

### 4.1. Participants

The participants of this study, 56 in all, were university students majoring in computer science and information management, aged from 22 to 24 years. The experimental and control groups were divided into two groups of 28, with 13 men and 15 women in each group.

### 4.2. Learning Process

This study designed an English reading SRL system using eye-tracking technology, through which the student's gaze position and dwell time were monitored. This was to analyze their emotions throughout the learning process and to provide them with SRL indicators after learning, thereby allowing them to self-check their learning situation, which would help them achieve SRL. Prior to the experiment, the senior English teachers and digital learning experts of the school's language center discussed the experimental materials and questionnaires and designed the test for the reading materials. The learning process is illustrated in Figure 8.

The experimental processes are as follows:

1. Pre-learning assignments

A 30-min pre-test and a 10-min pre-questionnaire were administered to the students (learning motivation, self-efficacy, and learning anxiety). Prior to the start of the learning activity, the student system was trained on how to operate.

2. Learning activities

The learning activities were conducted in a computer classroom, over the course of three weeks. The learning duration of each session was set to 20 min, and each group of students used the learning system to complete the English reading. The control group students used a multimedia-learning system with eye-tracking technology and no SRL mechanism. Nevertheless, the current reading recorded the students' dwell times through the eye tracker; when the students completed the reading, the quiz function allowed them to review their learning outcomes. By contrast, the students in the experimental group used a SRL mechanism with eye-tracking technology, they set their reading goals before learning, and used the marking function provided by the system, to apply the 5W1H-reading strategy, to determine the article character (who), event time (when), location (where), the article event (what), the purpose of the article (why), and the article conclusion (how). When the experimental group completed the reading, the system provided the quiz function, which allowed the students to view their self-learning results. This was completed to motivate them to focus on the materials and to assist them in the SRL strategy refinement. Moreover, the quiz function provided them with answers through instant feedback.

In this way, students in the experimental group completed SRL, and the system provided them with indicators to self-check their learning status. This study compares (i) the learning outcomes from using the SRL mechanism with eye-tracking technology, (ii) the dwell times and attention of the experimental and the control groups, (iii) the relationship between the students' learning outcomes with the emotion recognition technology, to analyze their attention and eye movement behavior. It also explores whether the students' attention and dwell time affect their SRL outcomes by using the SRL mechanism with eye-tracking technology.

3. Post-test

Once each group completed the learning activity, the students performed a 30-min post-test and a 10-min post-questionnaire (learning motivation, self-efficacy, learning anxiety, cognitive load, and technology acceptance).

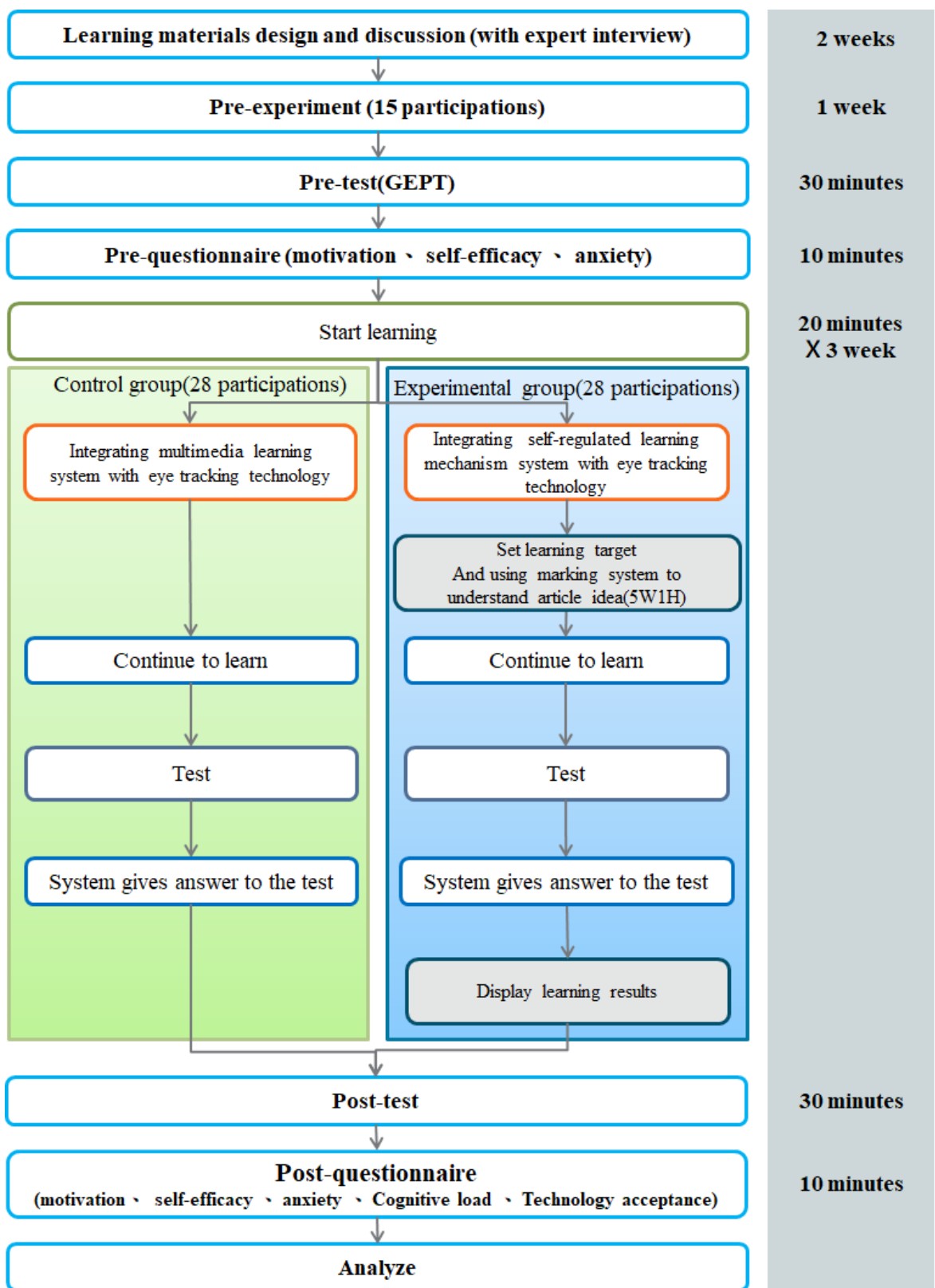

**Figure 8.** Comparison of the e-learning process between the experimental group and the control group.

*4.3. Research Tools and Analysis Methods*

4.3.1. Learning Materials and Tests

In this study, senior English teachers at the University Language Center guided the teaching methods and experience and jointly developed the reading materials. According to the experts' suggestions, the selected reading material belongs to the second level of the "In Focus" series of books published by Cambridge University Press and is used by intermediate reading courses in colleges and universities. Experts recommend that courses be arranged in this experimental system.

The learning outcome test was divided into two parts: the pre-test and post-test.

(i)    Pre-test

In order to understand the students' level in English reading prior to SRL, topics related to the reading ability test were selected from the mock test questions provided by the official website of the General English Proficiency Test (GEPT). The quiz questions were answered using single-choice questions. The content comprised 20 questions (100%), with an average of five points for each question, making up a total score of 100 points.

(ii)   Post-test

The post-test was based on the English reading materials in the learning activities, and the test questions related to the reading materials were compiled to analyze the difference in the learning outcomes between the experimental and control groups, using the SRL mechanism. A total of 10 questions (100%), with an average of 10 points and a total score of 100 points, were included.

4.3.2. Learning Motivation Questionnaire

The questionnaire is an important part of this study. The presence of multiple learning factors may have different influences on the learning state, with various interactive influences between each factor. Therefore, the design of the experiment and an appropriate analytical method were key to determining whether accurate experimental results could be obtained. Therefore, this study specifically was aimed at each learning factor and asked the experts to design appropriate questionnaires and to conduct experiments with the experimental design. The analysis method of the questionnaire was to carry out individual analyses for the different learning influencing factors. To avoid interaction among the different factors, the covariate analysis method was adopted.

First, this questionnaire was cited from the proposed motivation questionnaire. The questionnaire was suggested and confirmed by English experts and digital learning experts. One of the main aims of this study was to discuss whether there were significant changes in the English reading motivation before and after the e-learning activities. The subjects of the pre- and post-questionnaires were the same, with a total of four questions. A 5-point Likert scale was used, and the estimated value of Cronbach's alpha was 0.91.

4.3.3. The Learning Status Questionnaire

The learning status questionnaire contained five types of questions:

(i)    Self-efficacy questionnaire

The self-efficacy questionnaire was adapted from the personal self-efficacy questionnaire, the content of which has been confirmed by the experts. This paper was aimed at discussing the learning performance of students before and after their personal learning. Four questions were asked in the pre- and post-questionnaire stages. A 5-point Likert scale was used, and Cronbach's alpha calculated before and after the questionnaire was 0.93.

(ii)   Learning anxiety questionnaire

The foreign language learning anxiety questionnaire was adapted from the foreign language learning anxiety questionnaire, which primarily discusses the changes in learning

anxiety before and after learning. Four questions were asked in both the pre- and post-questionnaire stages. A 5-point Likert scale was used, and Cronbach's alpha calculated before and after the questionnaire was 0.93.

(iii)   Cognitive load questionnaire

The cognitive load questionnaire was adapted from the proposed cognitive load questionnaire. The questionnaire was divided into two parts: mental load and effort. The mental load study judged whether the difficulty of the e-learning material was perceived as a load by the students during the learning process; a total of three questions were asked. Cronbach's alpha, calculated using a 7-point Likert scale, was 0.86. Meanwhile, mental effort refers to when the brain working hard to explore the content of the e-learning materials used in the activities, which places a load on the student's learning process; two questions were asked. The 7-point Likert scale resulted in a Cronbach's alpha of 0.85.

(iv)   Technology acceptance questionnaire

The acceptance of science and technology is quoted from the questionnaires proposed. To explore whether the system function is helpful for students and whether the system is easy to use, a total of five questions were asked. The 5-point Likert scale results indicated a Cronbach's alpha of 0.94.

(v)   Interview items

The interview questionnaire cited was adapted for integrating technology into teaching interviews; it contained a total of four questions. Upon the completion of the experimental activities, individual interviews were conducted to learn more about the students' ideas and suggestions regarding the learning activities.

### 4.3.4. Self-Regulated Value

Under the SRL mechanism for developing the online learning system, teachers can examine the learning process of students and check their learning status. Students can review their own learning records and adjust their learning strategies. Furthermore, based on the data recorded by teachers and students using the computer-assisted SRL model, which developed the concept of self-regulated scores, students' self-regulated scores were divided into three levels: confidence, regularity, and spontaneity. Each facet used different parameters to obtain a student's self-regulated score. This study refers to the concept of self-regulated score calculations, with a self-regulated value, which was divided into two aspects: confidence and diligence. Confidence uses the student's time target achievement rate and attention target achievement rate; diligence was calculated using the student's marking scores and the unit test scores, as shown in Table 2.

**Table 2.** Aspects and parameters in the self-regulated value.

|  | **Aspect** | **Parameter** |
|---|---|---|
| Self-regulated value | Confidence | Time target achievement rate |
|  |  | Attention target achievement rate |
|  | Diligence | Marking scores |
|  |  | Unit test scores |

*(i)*   Confidence:

Confidence represents the students' expectations before learning. It takes into account the concept of the confidence factor analysis proposed in a prior study [22] that corresponds to the students setting their targets before learning. Learning attention and time are the confidence and expectations of this study. Moreover, the degree of attention is divided into

ten levels (1–10), such that "*j*" represents the number of times that learning is performed (in this experiment, *j* = 3), with the following formula:

$$Confidence(S_i)_j = \frac{Achievement\_Time(S_i)_j + Achievement\_Attention(S_i)_j}{2} \times 10. \quad (1)$$

In the above formula,

$$Achievement\_Time(S_i)_j = 1 - \frac{Actual\_Time(S_i)_j}{Setting\_Time(S_i)_j} \quad (2)$$

and

$$Achievement\_Attention(S_i)_j = \frac{Actual\_Attention(S_i)_j}{Setting\_Attention(S_i)_j} - 1 \quad (3)$$

Among these, $Achievement\_Time(S_i)_j$ represents the time achievement rate of student $S_i$ in the j-th study, $Achievement\_Attention(S_i)_j$ represents the achievement rate of the *i*-th student's attention target in the *j*-th study, and $Setting\_Time(S_i)_j$ and $Setting\_Attention(S_i)_j$ represent the students $S_i$'s expected learning time and attention before the *j*-th study, respectively. Following the *j*-th learning of student $S_i$, the $Actual\_Time(S_i)_j$ and $Actual\_Attention(S_i)_j$ represent the actual learning time and learning attention, respectively.

*(ii)* Diligence:

Diligence represents a student's efforts to learn. Based on the self-regulated aspect proposed by Thérèse, the self-regulated score is the sum of the scores for cognition, strategy, motivation, and achievement. It is calculated as follows:

$$Diligence(S_i)_j = \frac{Mark(S_i)_j + Score(S_i)_j}{2}. \quad (4)$$

where $Mark(S_i)_j$ represents the marking score (1–10 points) of student $S_i$ in the j-th learning, $Score(S_i)_j$ represents the unit test score of the student $S_i$ in the j-th learning (1–10 points), and $Confusion(S_i)_j$ represents the degree of confusion of the student $S_i$ in the system judgment during the j-th learning session (1–10 points).

Finally, after obtaining the values of the two facets, a self-regulated value can be obtained.

$$SelfRegularity\_Score(S_i)_j = \frac{Confidence(S_i)_j + Diligence(S_i)_j}{2} \quad (5)$$

4.3.5. Analysis Methods

This study used an independent sample *t*-test, Pearson's product-moment correlation coefficient analysis, and a covariate analysis (ANCOVA) to analyze the results. Specifically, the pre-tests were performed for the students in the experimental and control groups. The results were tested with independent samples, to check whether there was a significant difference between the students' prior English knowledge. At the end of the experiment, two groups of students were tested while reading English. In the absence of significant differences in the content of the pre-test, the ANCOVA was used to test whether there were significant differences in the post-test scores between the two groups, to check if the use of SRL mechanisms affected the students' learning outcomes. Additionally, Pearson's plot correlation analysis was used to analyze the experimental group, and the correlation between the degree of attention, confusion, marked score, and learning performance was analyzed to explore the relationship between the variables. Finally, an independent sample *t*-test was used to investigate the learning motivation, self-efficacy, and learning anxiety of the two groups of students before learning. The ANCOVA test was used to investigate

whether there were significant differences between the two groups of students before and after the study.

## 5. Experimental Results

### *5.1. Learning Performance*

5.1.1. Pre-Test

To analyze whether the two groups of students had the same prior knowledge of English reading, they were pre-tested before being allowed to read, as part of the formal experimental activities. The differences in English reading ability between the two groups of students were analyzed, using an independent sample *t*-test. As shown in Table 3, the average pre-test scores of the two groups of students did not reach a significant difference ($t = -0.57$, $p = 0.57 > 0.05$), indicating no significant difference in English reading ability between the two groups before the experiment.

Further sex-differentiated samples were used for independent sample *t*-tests. No significant differences were observed in the English reading abilities of either males or females before the experiment (male $t = -0.31$, $p = 0.76 > 0.05$; female $t = -0.51$, $p = 0.61 > 0.05$).

**Table 3.** Pre-test of the students between the experimental group and the control group.

| Group | N | Mean | SD | t |
|---|---|---|---|---|
| Experimental | 28 | 74.46 | 8.75 | $-0.57$ |
| Control | 28 | 75.71 | 7.54 | |
| Males in the experimental group | 13 | 74.23 | 10.58 | $-0.31$ |
| Males in the control group | 13 | 75.39 | 8.28 | |
| Females in the experimental group | 15 | 74.67 | 7.19 | $-0.51$ |
| Females in the control group | 15 | 76.00 | 7.12 | |

5.1.2. Post-Test

To analyze the learning performance of students after the completion of learning activities, a subsequent post-test was conducted. The results of the pretest were subjected to covariate analysis. To meet the hypothesis of this covariate analysis, the homogeneity test of variance was performed with two groups of results. The significance of the test was 0.66, which did not reach a significant level, indicating that the variance of the test scores of both groups was homogenous. Following the analysis of homogeneity of regression coefficients, whether there existed a SRL mechanism after the experimental and the control group did not reach a significant level ($F = 0.75$, $p = 0.08 > 0.05$), indicating there was no interaction between the two groups. Homogeneity, in line with the basic assumptions of covariate analysis, allows for ANCOVA verification.

The ANCOVA results are listed in Table 4. The post-test results showed that the learning performance of the experimental group was significantly higher than that of the control group ($F = 12.35$, $p = 0.001 < 0.05$). From this result, it was discerned that the students in the experimental group achieved better results than those in the control group.

**Table 4.** Post-test of students between the experimental group and the control group.

| Group | N | Mean | SD | Adjusted Mean | F | $\eta^2$ |
|---|---|---|---|---|---|---|
| Experimental | 28 | 83.93 | 9.94 | 84.23 | 12.35 ** | 0.19 |
| Control | 28 | 70.71 | 18.84 | 70.41 | | |

** $p < 0.001$.

Further sex-differentiated samples were used for independent sample *t*-tests, the results of which are presented in Table 5. The results indicate that the learning effectiveness of males in the experimental group was significantly higher than that in the control group ($t = 3.64$, $p = 0.001 < 0.05$). In other words, the system can help male students enhance

learning performance; however, the learning performance of the two groups of females did not show significant differences (t = 0.90, *p* = 0.38 > 0.05), indicating the system had no significant effect on the learning performance of females.

**Table 5.** Post-test of students of different genders between the experimental group and the control group.

| Group | N | Mean | SD | t | d |
|---|---|---|---|---|---|
| Males in the experimental group | 13 | 90.00 | 9.13 | 3.64 * | 1.43 |
| Males in the control group | 13 | 66.15 | 21.81 | | |
| Females in the experimental group | 15 | 78.67 | 7.43 | 0.90 | - |
| Females in the control group | 15 | 74.67 | 15.52 | | |

* *p* < 0.05.

### 5.2. Analysis for the SRL Mechanism

#### 5.2.1. Attention

In this study, the dwell time was used to define the attention level, and an independent sample *t*-test was used to check if there was a difference in attention between the students of the experimental and the control groups, who used the SRL mechanism. As shown in Table 6, the average attention level of the two groups exhibited a significant level of difference (t = 2.81, *p* = 0.007 < 0.05); that is, the SRL mechanism developed in this study significantly improved the students' attention. Furthermore, an independent sample *t*-test was conducted for the gender differences, which revealed that the attention levels of the males in the two groups differed significantly (t = 3.75, *p* = 0.001 < 0.05). This implies that the developed SRL mechanism system has significant benefits for male students. However, the attention levels of the females in the two groups did not significantly differ (t = 0.58, *p* = 0.57 > 0.05), revealing that the SRL mechanism has no significant benefit for female students.

**Table 6.** Attention of students between the experimental group and the control group.

| Group | N | Mean | SD | t | d |
|---|---|---|---|---|---|
| Experimental | 28 | 7.93 | 0.90 | 2.81 * | 0.75 |
| Control | 28 | 6.96 | 1.58 | | |
| Males in the experimental group | 13 | 8.15 | 0.69 | 3.75 * | 1.46 |
| Males in the control group | 13 | 6.39 | 1.56 | | |
| Females in the experimental group | 15 | 7.73 | 1.03 | 0.58 | 0.21 |
| Females in the control group | 15 | 7.47 | 1.46 | | |

* *p* < 0.05.

#### 5.2.2. Marking Score

Furthermore, we analyzed the marking function status of the different gender students in the experimental group. When the students marked the correct sentence (5W1H), the system output points in the range 1–10, and the scores of the different genders in the experimental group were scored independently. The sample *t*-test results are listed in Table 7. The marking scores of the male and female students showed significant differences (t = 2.12, *p* = 0.04 < 0.05). The results showed that male students had significantly better use of the marking function than female students.

**Table 7.** Marking score between the different genders.

| Group | N | Mean | SD | t | d |
|---|---|---|---|---|---|
| Male | 13 | 7.62 | 1.12 | 2.12 * | 0.81 |
| Female | 15 | 6.67 | 1.23 | | |

* *p* < 0.05.

### 5.3. Correlation Analysis of the SRL Mechanism and the Learning Performance

In the SRL English reading environment, the learning performance of the subjects showed a high positive correlation with the average learning attention (r = 0.53, $p$ = 0.004 < 0.05) and the average marking score (r = 0.44, $p$ = 0.019 < 0.05). The results presented above in Table 8 show that the students of the experimental group, using the SRL mechanism of the English reading system, had a positive influence on the learning outcomes in the learning process. However, the degree of confusion negatively affects the learning performance. Students with a high attention and low confusion achieved good results in their learning performance.

**Table 8.** Correlation analysis of the students' self-regulated learning and the learning performance in the experimental group.

|  | Post-Test | | |
|---|---|---|---|
|  | **Pearson Correlation** | **Significant (Two-Tailed)** | **N** |
| Post-test | 1 |  | 28 |
| Attention | 0.53 | 0.004 * | 28 |
| Marking score | 0.44 | 0.019 * | 28 |

* $p$ < 0.05.

### 5.4. Questionnaire Analysis

In this section, the individual analysis of various types of questions from the questionnaire, is performed in consideration of the different learning influencing factors. In the analysis process, to avoid the interaction among different learning influencing factors, the covariate analysis method was adopted to solve this problem of collinearity of the possible interactions among the different variable factors. The following analysis of each learning factor is divided into pre-questionnaire and post-questionnaire parts. The pre-questionnaire part proves there was no essential difference between the experimental and control groups before the experiment, to verify the fairness and objectivity of the grouping. In this study, the post-questionnaire analysis of the main five learning factors used the covariate analysis. We sorted the results of the pre- and post-questionnaires analysis of the five main learning factors, as shown in Table 9 below:

**Table 9.** Summary table of the significant effects of the e-learning system using the SRL mechanism on the major learning factors.

|  | **Learning Motivation** | **Self-Efficacy** | **Learning Anxiety** | **Cognitive Load** | **Technology Acceptance** |
|---|---|---|---|---|---|
| Significant (Pre-questionnaire) | No | No | No | No | No |
| Significant (Post-questionnaire) | Yes | Yes | Yes | Yes | Yes |

#### 5.4.1. Learning Motivation

The motivation questionnaire revealed the motivation for learning English reading before and after the students' activities, using a 5-point Likert scale. The analysis of the motivation from the pre-questionnaire for English learning, used an independent sample $t$-test. It revealed, as listed in Table 10, no significant differences in the learning motivation between the two groups (t = 0.97, $p$ = 0.34 > 0.05, Cohen's d = 0.26), implying that the two groups had a similar learning motivation before the e-learning activities.

**Table 10.** Pre-questionnaire of the motivation of the students between the experimental group and the control group.

| Group | N | Mean | SD | t |
|---|---|---|---|---|
| Experimental | 28 | 3.68 | 0.60 | 0.97 |
| Control | 28 | 3.50 | 0.76 | |

The post-questionnaire of the motivation was analyzed using the ANCOVA, and the pre-learning motivation questionnaire was used as a covariate. According to the hypothesis of this analysis, the homogeneity test of variance was performed with the two groups of questionnaires with respect to the learning motivation. The significance of the test was 0.90, which failed to reach a significant level, thereby indicating that the two groups of the learning motivation questionnaires were homogenous. Subsequently, through the regression coefficient homogeneity analysis, it was observed that the results of the learning motivation questionnaires of the two groups did not reach a significant level ($F = 0.79$, $p = 0.38 > 0.05$). This implies that the two groups did not interact with each other. It is homogenous and conforms to the basic assumptions of the covariate analysis, so it can be used for the ANCOVA verification. The results are presented in Table 11. The learning motivation of the experimental group was significantly higher than that of the control group ($F = 14.77$, $p < 0.05$).

**Table 11.** Post-questionnaire of the motivation of the students between the experimental group and the control group.

| Group | N | Mean | SD | Adjusted Mean | F | η2 |
|---|---|---|---|---|---|---|
| Experimental | 28 | 4.26 | 0.63 | 4.21 | 14.77 * | 0.22 |
| Control | 28 | 3.71 | 0.49 | 3.76 | | |

* $p < 0.05$.

### 5.4.2. Self-Efficacy

Self-efficacy primarily discusses the students' learning efficiency before and after learning. A 5-point Likert scale was used, and Cronbach's alphas of the pre-and post-questionnaires were 0.86 and 0.85, respectively.

An independent sample *t*-test was used to analyze the self-efficacy pre-questionnaire. The results are listed in Table 12. No significant differences were observed in self-efficacy between the two groups before the experiment ($t = 0.41$, $p = 0.69 > 0.05$), indicating that the self-efficacy values of the two groups were comparable before the learning activity.

**Table 12.** Pre-questionnaire of the self-efficacy of the students between the experimental group and the control group.

| Group | N | Mean | SD | t |
|---|---|---|---|---|
| Experimental | 28 | 3.20 | 0.83 | 0.41 |
| Control | 28 | 3.11 | 0.81 | |

The post-questionnaire self-efficacy also used the ANCOVA test analysis. The corresponding questionnaire was used as a covariate to analyze the post-questionnaire through the ANCOVA. The significance of the homogeneity test of variance number was 0.27, which did not reach a significant level, thereby implying that the two groups were homogenous. The regression coefficient homogeneity test also did not reach a significant level ($F = 0.28$, $p = 0.60 > 0.05$), indicating that the two groups did not interact and were homogenous, which met the basic assumptions of the covariate analysis. The results of this analysis are presented in Table 13 and show that the self-efficacy of the experimental group was significantly higher than that of the control group ($F = 25.89$, $p < 0.05$).

**Table 13.** Post-questionnaire of the self-efficacy of the students between the experimental group and the control group.

| Group | N | Mean | SD | Adjusted Mean | F | η2 |
|---|---|---|---|---|---|---|
| Experimental | 28 | 4.18 | 0.72 | 4.16 | 25.57 ** | 0.33 |
| Control | 28 | 3.45 | 0.49 | 3.47 | | |

** $p < 0.01$.

### 5.4.3. Learning Anxiety

Foreign language learning anxiety mainly explores learning changes in students before and after learning, using a 5-point Likert scale. Prior to and after the administration of the questionnaire, Cronbach's alphas were 0.72 and 0.80, respectively.

The analysis of the anxiety pre-questionnaire for foreign language e-learning used an independent sample *t*-test. The results showed that there was no significant difference between the two groups (t = 0.36, $p = 0.72 > 0.05$), which are shown in Table 14.

**Table 14.** Pre-questionnaire of the learning anxiety of the students between the experimental group and the control group.

| Group | N | Mean | SD | t |
|---|---|---|---|---|
| Experimental | 28 | 3.55 | 0.78 | 0.36 |
| Control | 28 | 3.47 | 0.70 | |

Furthermore, the ANCOVA test showed that learning anxiety in the experimental group was significantly lower than that in the control group. The significance of the homogeneity test of the variance number was 0.55, which did not reach a significant level, implying that the two groups were homogenous. The homogeneity test of the regression coefficient also did not reach a significant level (F = 0.06, $p = 0.80 > 0.05$), which indicates that the two groups did not interact and were homogenous; this met the basic assumptions of the covariate analysis. The results of the ANCOVA analysis are shown in Table 15 (F = 10.99, $p < 0.05$).

**Table 15.** Post-questionnaire of the self-efficacy of the students between the experimental group and the control group.

| Group | N | Mean | SD | Adjusted Mean | F | η2 |
|---|---|---|---|---|---|---|
| Experimental | 28 | 2.75 | 0.74 | 2.73 | 10.99 ** | 0.17 |
| Control | 28 | 3.29 | 0.75 | 3.30 | | |

** $p < 0.01$.

### 5.4.4. Cognitive Load

For the cognitive load, three questions were asked and a 7-point Likert scale was used. For the mental effort, two questions were asked and a 7-point Likert scale was used. Cronbach's alpha was 0.81. An independent sample *t*-test analysis was performed, and the results are listed in Table 16. The two groups were found to exhibit significant differences in the mental workload (t = −3.27, $p = 0.002 < 0.05$); moreover, the mental effort showed a significant difference (t = −2.39, $p = 0.02 < 0.05$). These results imply that although the two sets of learning materials were the same, the proposed SRL mechanism helped the students understand the content of the article, thereby reducing the mental workload and effort. Therefore, the experimental results show that the experimental group using the e-learning system with the SRL mechanism can significantly reduce the learning anxiety of e-students in the reading learning processes, compared with the control group.

**Table 16.** Experimental group and control group cognitive load's independent sample *t*-test results.

| Group | Group | N | Mean | SD | t | d |
|---|---|---|---|---|---|---|
| Mental workload | Experimental | 28 | 2.54 | 0.95 | −3.27 * | −0.86 |
|  | Control | 28 | 3.39 | 1.02 |  |  |
| Mental effort | Experimental | 28 | 2.46 | 1.21 | −2.39 * | −0.64 |
|  | Control | 28 | 3.29 | 1.36 |  |  |

\* $p < 0.05$.

### 5.4.5. Technology Acceptance

The technology acceptance questionnaire was rated on a 5-point Likert scale, and Cronbach's alpha was found to be 0.87. An independent sample *t*-test was performed. The results are presented in Table 17 and indicate that the acceptance of the experimental group for the English reading system using the SRL mechanism developed in this study, was significantly better than that of the control group (t = 4.08, $p < 0.05$). According to the interview results, students who used the SRL-based English reading system felt that the methods provided by the system could help them correct their learning strategies and effectively improve their learning outcomes, thus making the technology acceptance significantly higher than that of the control group students.

**Table 17.** Experimental group and control group technology acceptance questionnaire's independent sample *t*-test results.

| Group | N | Mean | SD | t | d |
|---|---|---|---|---|---|
| Experimental | 28 | 4.66 | 0.39 | 4.08 * | 1.09 |
| Control | 28 | 4.08 | 0.64 |  |  |

\* $p < 0.05$.

## 6. Discussion

In the case of the students using the self-regulated English reading mechanism, the learning performance of the experimental group was significantly better than that of the control group. In the case where the experimental and control groups, the students had the same prior knowledge of English reading, the post-test scores of the experimental group were higher than those of the control group, as displayed in the English reading study course. The SRL mechanism integrated with eye-tracking technology in the English reading system, in this study, did improve the learning outcomes.

In the case of students using the self-regulated English reading mechanism, the learning performance of the male students in the experimental group was significantly better than that of those in the control group.

The results indicate that gender differences can affect the learning performance. The learning performance of the male students in the experimental and control groups was significantly different. However, the learning performance between the two groups' female students was not significantly different; the interviews revealed similar results. Female students believe that the system provides many features for marking, including the quiz function during the learning process, which, in turn, affects their learning strategies. It can be inferred that the SRL mechanism integrated with eye-tracking technology in the English reading system, developed in this study, is more effective in improving the learning performance of male students.

In the case of students using the self-regulated English reading mechanism, the experimental group students' attention was significantly higher than that of the control group students, and the confusion of the former was significantly lower than that of the latter.

The results show that in the English reading-learning system, the learning attention of the experimental group students was significantly better than that of the control group. Conversely, in terms of the decrease in confusion, the experimental group exhibited significantly less confusion than the control group. Therefore, the use of the proposed SRL

mechanism for English reading systems has significant benefits in improving the attention and reducing confusion during learning.

The attention and marking scores of the experimental group students were positively correlated with the learning performance, whereas confusion was negatively correlated. Furthermore, in the proposed SRL mechanism for English reading, the average attention and marking scores of the experimental group students were positively correlated with the students' learning performance and negatively correlated with the average confusion. This indicates that the higher the students' attention and marking score, the lower their learning confusion and the better their learning performance.

Additionally, the students' learning motivation and self-efficacy were significantly improved and their learning anxiety was reduced after using the SRL mechanism integrated with eye-tracking technology in English reading.

A comparison of the results of the questionnaires administered before and after the experiment indicated that in the English reading and learning system, the learning motivation and self-efficacy of the experimental group were significantly better, and the learning anxiety was significantly lower after the experiment. By contrast, the experimental group exhibited a greater learning motivation and self-efficacy and a significantly lower learning anxiety than the control group. This result shows that using the proposed SRL mechanism in an English reading system can effectively improve the students' learning motivation and self-efficacy, while reducing their learning anxiety by achieving a satisfactory SRL effect.

## 7. Conclusions and Future Work

### 7.1. Conclusions

In this study, we developed a SRL mechanism integrated with eye-tracking technology for English reading. This learning system detected the students' dwell time and regression times during the learning process and identified their attention and confusion, using the Tobii Pro X2-30 eye tracker. The data thus obtained indicated whether students were paying attention to the learning materials or were confused during the learning. When the attention of the students was reduced or confusion occurred, the quiz function provided the students with a periodic self-evaluation to reflect on their own learning status and achieve an effective SRL process. This study explored whether English students can use SRL mechanisms in their English reading courses to influence their learning outcomes. The mechanism developed in this study can effectively enhance the learning attention and reduce confusion throughout the learning process. Additionally, it greatly improves the learning motivation, the learning interest and self-efficacy, and reduces the learning anxiety throughout the learning process. Furthermore, an effective and good e-learning mechanism and learning method are proposed so as to achieve the sustainability of education by improving digital learning.

### 7.2. Recommendations for Future Research

The SRL system developed in this study proved that the learning results of the experimental group of students were significantly better than those of the control group of students, although the observations were only limited to learning through English reading. The marking function developed by this system can be applied to other language learning applications and to historical subjects that require clarification in terms of time, place, and characters. Based on the student interviews, a new feature with no specified clues can be added to help students make notes. In the future, brainwaves can be used. Consequently, whether the students have continued to focus on the e-learning materials, can be monitored and the relationship between attention and confusion can be compared, using the brain wave value, eye gaze time, and regression times.

**Author Contributions:** Conceptualization, Y.-C.K.; Methodology, Y.-C.K. and C.-B.Y.; Formal analysis, C.-B.Y.; Investigation, C.-Y.W.; Resources, Y.-C.K.; Data curation, Y.-C.K., C.-B.Y. and C.-Y.W.; Writing—original draft, C.-B.Y. and C.-Y.W.; Writing—review & editing, C.-B.Y.; Project administration, C.-B.Y. and C.-Y.W. All authors have read and agreed to the published version of the manuscript.

**Funding:** This study is supported in part by the National Science and Technology Council of the Republic of China under contract number MOST 111-2410-H-031-024.

**Institutional Review Board Statement:** Not applicable.

**Informed Consent Statement:** Not applicable.

**Data Availability Statement:** Not applicable.

**Conflicts of Interest:** The authors declare no conflict of interest.

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
