# Peer review of "A Strategy for Enhancing English Learning Achievement, Based on the Eye-Tracking Technology with Self-Regulated Learning"

_sustainability, doi:10.3390/su142316286_

Round 1

Reviewer 1 Report

From my perspective the article is excellent and I would like to strongly urge you to continue the research, because you have an excellent material here. My only suggestion, which is strictly about the look of the article, is to have the Figure 2 and 6 made clearer. It is only a suggestion.

Author Response

Please see the attachment. Thank you~

Reviewer 2 Report

This study reports on what an experiment designed to assess the pedagogical usefulness of a Self-Regulated Learning mechanism that uses eye-tracking technology. This is an original idea, which could have significant impact on the design of learning technology, but the way in which it is implemented seems like an attempt to find an ex post facto justification for a technology. 

A fundemental weakness of the article is that it seems to rest of a very nebumous conceptulisation of 'reading'. Reading is more than cursory movement of the eyes along a series of written signs - it involves various forms of engagement, from skimming and scanning to the active interrogation of the text. Not only is this conceptual complexity not accounted for in the  design of the experiment, but references to the literature on reading in an L2 are scarce and/or inapproriate.

The study also uses a large variety of psychological constructs as potential dependent variables - presumably in the belief that the more one tries, the likelier it is that some measures will cross the lowest threshold of statistical significance. The expansive discussion of attention, emotion recognition and language learning stragegies does not meaningfully connect to the literature: for instance, there is surprisingly little mention of the seminal work of scholars like Rebecca Oxford or Andrew Cohen in the discussion of strategies. 

In terms of writing, the paper seems to lack direction and expositional strength, which makes it hard to sustain reader interest. Overall, this is a paper that would require considerable revision, both conceptually and in terms of style before it reaches publication standards. 

Author Response

Please see the attachment. Thank you very much~

Reviewer 3 Report

Review. Integrating self-regulated learning mechanism and tracking technology for enhanced English reading

There are a number of basic flaws in the format and structure of this article that make it difficult to evaluate and review. Firstly, the tradition in academic papers is that they are presented in the past tense and this paper is presented throughout in the future tense. This leads the reviewer to suspect that elements of the paper have been extracted from a previous research proposal. 

Secondly the use of reference followed by a number makes it clumsy and difficult to read. There are various forms of wording that could be used here, ‘research has shown (5) that etc. 

There is no clear demarcation of the methodology and participants, and it seems that the reader must have processed the majority of the paper before the appropriate format of results and discussion are introduced. This means that regardless of the quality of the research itself, the paper as it currently stands must be rejected and a full rewrite requested. The abstract itself requires attention, because the system has not been proposed but evaluated in this paper, and details on the participants, the methodology and the results should be included in the abstract, so that readers could repeat the experiment themselves.  

I also don’t think that the authors can claim the methodology 5W1H as their own, this is a well-know approach used internationally to structure reading comprehension.

The approach adopted here seems to be to create a self-regulated learning platform for participants to monitor their own progress using feedback on their eye-movements to help to monitor their attention, and thereby to increase their motivation and feelings of self-efficacy.  The authors have identified key elements here emerging in the research, the importance of self-confidence in motivation and performance, and the importance of eye movements, which as the authors suggest have been largely overlooked in recent research in reading. The system was evaluated with 28 experimental participants and 28 controls, who undertook equivalent tasks without the use of the self-regulating eye movement system.  The figures here are explanatory and suggest that the approach was well-designed and delivered, with appropriate post-tests and controls. 

The description of the questionnaires would be more appropriate under the Methods and Materials section of the paper. Participations should be changed to participants. 

Incomplete sentence p19 top. The main purpose…

In terms of presentation of the results, it would be more effective to cluster the questionnaire results into 1 table to report this which would allow the differences between the groups to be more easily followed. 

There seems to be an error in the discussion on the results of motivation studies, which were shown not to be significantly different between the 2 groups, either pre or post test, although they are later claimed to be significant.  Could the authors check this, and make it clearer which is correct.

In conclusion there is material worthy of publication here, but the conventions of academic writing need further attention.  The article should then prove an interesting contribution to the literature in the fi

Author Response

(The authors gave the same response as above.)

Reviewer 4 Report

Manuscript Sustainability- with title “Integrating Self-Regulated Learning Mechanism and Eye-Tracking Technology for Enhanced English Reading Performance”. The study makes it possible to analyze and capture learners' real-time learning status, and then provide timely learning reminders to help them achieve self-defined learning goals and effectively improve active learning interest and learning performance. Learner attention is an important part of recognition technology. The proposal presents a design with a self-regulated learning mechanism based on eye-tracking technology. The proposed learning system detects learners' eye movements during learning and identifies attention and fixation states. The goal is to help with an attention-enhancing mechanism; the mechanism could help learners maintain a better reading state, thus improving learning performance. The study explores students' learning motivation, motivation, self-efficacy, learning anxiety, and the performance of the SRL mechanism based on eye-tracking recognition technology. The results of this study show that learners who used an SRL mechanism based on eye-tracking technology had higher learning performance than those who did not use the system.

To improve its work, I suggest.

1.                  I suggest applying the magazine template located at https://www.mdpi.com/journal/sustainability/instructions

2.                  The Abstract should be between 150 and 200 words; please revise.

3.                  In the Abstract, include a paragraph about future work on your research.

4.                  Check the English grammar and spelling of the entire paper.

5.                  Improve the quality and resolution of Figure 1.

6.                  Improve the quality and resolution of Figure 5.

7.                  Improve the quality and resolution of Figure 8.

8.                  Reinforce the explanation of the method used in the literature review.

9.                  Reinforce the explanation Discussion.

10.              Strengthen the Discussion section.

11.              For your research to have more support with all the defined tables and summaries, I suggest that you place the study data and tables in a dataset; you can create one at https://data.mendeley.com/

12.              I suggest updating the references; 90% of the 43 references do not correspond to the five years of validity. 

Author Response

(The authors gave the same response as above.)

Reviewer 5 Report

I'd want to start off by saying that the pertinent study is valuable and a contribution to the field.

With this in mind;

1-I'd like to point out that there are certain APA-related spelling and referencing issues.

2. I believe the text requires editing.

3- I believe it is vital to explain more clearly why there is no gender difference.

4- The study should be supported by more recent sources.

Author Response

 "Please see the attachment." Thank you very much~ 

Round 2

Reviewer 2 Report

I acknowledge the authors’ response, and maintain that the paper suffers from fundamental limitations that cannot be addressed through surface-level editing. Calling a method “rigorous” in editorial correspondence  is not a sufficient improvement for making it appropriate; the additional reference to the article by Cohen does not improve the literature review sufficiently so as to warrant the study. 

Author Response

Please see the attachment. Thank you~ 

Reviewer 3 Report

I would like the authors for their attempts to clarify the paper in order to reach the standards necessary for publication.  There is no doubt that the material and research approach are now much clearer than previously.  However, there is still extensive work to complete to make the English appropriate for an academic journal. 

Many of the sentences do not make sense as they are currently written, too many to go through and itemise.  These of scholar Cohen is polite but not necessary in an article of this time, a simple Cohen is  more appropriate.  A number of sentences place the definite artic (the) in the wrong place.  The article would really benefit from input from a native English speaker, to help clarify the research undertaken.  

In conclusion the article has significantly improved, but still does not meet the standards for publication.

Author Response

(The authors gave the same response as above.)

Round 3

Reviewer 2 Report

I acknowledge the authors’ response, and maintain that the paper suffers from fundamental limitations that cannot be addressed through surface-level editing. This is a conceptually flawed and methodologically weak paper that does not have potential to add to the scholarly record. 

Author Response

"Please see the attachment." Thank you~ 

Reviewer 3 Report

Dear authors, I would like to thank you for your efforts to address my remaining concerns with this article, which in my view is now (almost) ready for publication.  I note however, that there are a couple of stray references at the end of your list that should be incorporated into the text or removed.

Author Response

(The authors gave the same response as above.)
